# Design and Evaluation of Engineered Extracellular Vesicle (EV)-Based Targeting for EGFR-Overexpressing Tumor Cells Using Monobody Display

**DOI:** 10.3390/bioengineering9020056

**Published:** 2022-01-29

**Authors:** Hiroaki Komuro, Shakhlo Aminova, Katherine Lauro, Daniel Woldring, Masako Harada

**Affiliations:** 1Institute for Quantitative Health Science and Engineering (IQ), Michigan State University, East Lansing, MI 48824, USA; komurohi@msu.edu (H.K.); aminovas@msu.edu (S.A.); lauroka1@msu.edu (K.L.); woldring@msu.edu (D.W.); 2Department of Biomedical Engineering, Michigan State University, East Lansing, MI 48824, USA; 3Department of Chemical Engineering and Materials Science, Michigan State University, East Lansing, MI 48824, USA

**Keywords:** extracellular vesicles, EV engineering, EGFR targeting

## Abstract

Background: Extracellular vesicles (EVs) are attracting interest as a new class of drug delivery vehicles due to their intrinsic nature of biomolecular transport in the body. We previously demonstrated that EV surface modification with tissue-specific molecules accomplished targeted EV-mediated DNA delivery. Methods: Here, we describe reliable methods for (i) generating EGFR tumor-targeting EVs via the display of high-affinity monobodies and (ii) in vitro measurement of EV binding using fluorescence and bioluminescence labeling. Monobodies are a well-suited class of small (10 kDa) non-antibody scaffolds derived from the human fibronectin type III (FN3) domain. Results: The recombinant protein consists of the EGFR-targeting monobody fused to the EV-binding domain of lactadherin (C1C2), enabling the monobody displayed on the surface of the EVs. In addition, the use of bioluminescence or fluorescence molecules on the EV surface allows for the assessment of EV binding to the target cells. Conclusions: In this paper, we describe methods of EV engineering to generate targeted delivery vehicles using monobodies that will have diverse applications to furnish future EV therapeutic development, including qualitative and quantitative in vitro evaluation for their binding capacity.

## 1. Introduction

The last 20 years have transformed understanding of extracellular vesicle (EV) biology from early indications that EVs were waste disposal machinery into a realization that EVs are vehicles for intercellular communication [1]. Cells naturally secrete heterogeneous populations of lipid bilayer membranous nanoparticles termed exosomes and microvesicles (MVs). Their composition is affected by several factors, such as cellular response to the physiological and pathological condition or surrounding tissues affects their composition [2]. While the size range largely overlaps, the distinctive biogenesis differentiates exosomes and MVs. Exosomes (40–150 nm in diameter) are derived from intraluminal vesicles of multivesicular bodies (MVB) and are released from the cell upon MVB-cell membrane fusion. MVs (50–1000 nm in diameter) are generally larger vesicles and are the product of direct budding from the plasma membrane [3,4]. EVs have been detected in all the body fluids, such as saliva, tears, urine, breast milk, blood serum, cerebrospinal fluids, and amniotic fluids [5,6,7,8], and are secreted by all kinds of cells to transport biomolecular cargo for cellular communication in both physiological and pathological conditions [9,10,11,12,13,14,15,16,17,18]. EV cargos are heterogeneous in composition, reflecting both the cell type of origin and status of the cells or surrounding tissues [2]. EV-mediated therapeutic delivery circumvents significant issues associated with synthetic nanoparticles, such as instability, immunogenicity, toxicity, and biological barrier crossing [1,19,20,21,22], making them an attractive alternative therapeutic carrier.

Our previous work described the EV surface protein engineering method [23] using protein fusion with a C1C2 domain of human lactadherin (milk fat globule EGF factor VIII/MFG-E8) fused with a pancreas-specific peptide [23,24,25]. We engineered HEK293T-derived EVs to target pancreatic cells in vitro and in vivo and demonstrated pancreas targeting and delivery of cargo DNA. EV surface protein modification enables EVs to display a protein with a high affinity towards a specific cell type [23]. In this report, we used the same EV surface approach with the use of an engineered monobody (E626), a clinically approved binding molecule with high affinity towards EGFR tumors [26].

Monobodies are a well-suited class of small non-antibody scaffolds derived from the thermodynamically stable human fibronectin type III (FN3) domain. Additional hydrophilic mutations have been incorporated into this scaffold (Fn3HP) to improve processing and in vivo biodistribution [27,28,29]. Site-wise modifications of three solvent exposed loops (akin to antibody CDRs) on the hydrophilic monobody scaffold has enabled strong, specific binding interactions against a diverse panel of clinically relevant targets [30].

EV surface display of engineered protein scaffolds allows for robust binding interactions between EVs and cellular or molecular targets of interest [20,31,32,33,34,35]. In the past, several studies demonstrated the EV targeting using surface engineering with lactadherin gene fusion, such as EGFR nanobody, carcinoembryonic antigen, and HER2 antibody [20,35,36,37,38]. It has been demonstrated that producer cells transfected with a gene fusion to lactadherin C1C2 plasmid displayed the coding gene on the EV surface [38,39,40].

In this study, we demonstrated a simple yet robust method to engineer EV surface molecules using lactadherin C1C2 fusion gene plasmid accompanied with comprehensive EV characterization and assays to test their binding capacity in vitro. Our study proposes a comprehensive procedure for EV surface engineering and characterization to harness the development of customizable EV-based targeted delivery vehicles for therapeutic EV engineering.

## 2. Materials and Methods

### 2.1. EV Monobody Display Plasmid Construction

The EV monobody display constructs were created using Seamless Ligation Cloning Extract (SLiCE) assembly, as previously described [23]. PCR fragment of EV display backbone from pcS-p88-C1C2 using the primer sets listed in Appendix A was fused with the synthetic double stranded DNA fragments coding for monobody (E626 or RDG) and G4S-PAS linker [41] including 15 bp overhangs. The three double-stranded DNA fragments were joined together by homologous recombination using SLiCE cloning. Fragment 1: PCR amplified product of EV display backbone amplified from pcS-p88-C1C2 with 15 bp overlaps at the HA and the G4S-PAS linker; Fragment 2: synthetic DNA of the monobody (E626 or RDG) with 15 bp overlaps with the end of HA tag and the start of the linker; Fragment 3: synthetic DNA of the linker that overlaps with both fragments 1 and 2 at each end. Figure 1A illustrates the EV monobody display construct assembly. All the synthetic DNAs were purchased from Integrated DNA Technologies (IDT; Coralville, IA, USA).

### 2.2. Cell Culture and Treatment

The following cell lines, obtained from American Type Culture Collection (ATCC), were tested for mycoplasma: HEK293T (Human Embryonic Kidney cell line), A431 (Human carcinoma cell line), and MCF-7 (human breast cancer cell line). The cells were cultured in high-glucose DMEM (Gibco) supplemented with 100 U/mL penicillin, 100 µg/mL streptomycin, and 10% (*v*/*v*) fetal bovine serum (FBS, Gibco). FBS was ultracentrifuged in PET Thin-Walled ultracentrifuge tubes (75000471, Thermo Scientific, Waltham, MA, USA) at 100,000 g with a Sorvall WX+ Ultracentrifuge equipped with an AH-629 rotor (*k* factor = 242.0) (Thermo Scientific, Waltham, MA, USA) for 18 h at 4 °C to remove the bovine EVs and create EV-depleted FBS for use in the culture media for preparation of engineered EVs. All cells were maintained in a humidified incubator with 5% CO_2_ at 37 °C. For EV production, EV display constructs were either transfected alone or along with an imaging EV display plasmid into HEK293T cells. In-house PEI (polyethylenimine, 408727 Sigma, St. Louis, MO, USA) transfection reagent was used, which works similarly to commercially available polymer- or liposome-mediated in vitro transfection reagents [42]. Cells were seeded at 2 × 10^6^ in a 10 cm cell culture dish for 24 h in regular culture media and transfected with 10 µg total DNA suspended with PEI in non-supplemented DMEM. To prepare the DNA-PEI transfection mixture, we added 10 µg DNA/100 mm dish to PEI in a ratio of 1:2.5 (DNA/PEI) in non-supplemented DMEM, pulse-vortexed for 30 s, and incubated at room temperature for 10 min. Following 24 h incubation, cells were washed twice with PBS, and the culture media was replaced with DMEM supplemented with EV-depleted FBS for another 24 h incubation for EV production. For naïve EV production, cells were cultured with DMEM supplemented with EV-depleted FBS without transfection for 24 h. E626-mCherry and RDG-mCherry co-labelled EVs were prepared from HEK293T cells transfected with the mCherry-EV display construct (pcDNA-mCherry-C1C2) and co-transfected with mCherry-EV display and E626-EV display constructs. EVs were labeled with gaussia luciferase (gLuc), with E626, or with RDG and were prepared by transfecting 293T with plasmid pcDNA-gLuc-C1C2 alone or with pcS-E626-C1C2 or pcS-RDG-C1C2 [43].

### 2.3. EV Isolation

The cells were grown in DMEM media supplemented with EV-depleted FBS for 24 h, and the media from the plates was collected. For each batch, EVs were purified from 20 mL of conditioned media by differential centrifugation. The media was centrifuged at 400 g for 10 min and to remove the cell and cell debris. In order to remove the contaminating apoptotic bodies, we centrifuged the media at 2000× *g* for 30 min. The supernatant was then ultracentrifuged in PET Thin-Walled ultracentrifuge tubes (Thermo Scientific 75000471) at 100,000× *g* with a Sorvall WX+ Ultracentrifuge equipped with an AH-629 rotor (*k* factor = 242.0) for 90 min at 4 °C to pellet the EVs [44]. The pellet containing EVs was resuspended in 100 µL PBS, except for the gLuc-labeled EVs, which were resuspended in DPBS (Gibco 14190136, Thermo Fisher Scientific, Waltham, MA, USA).

### 2.4. Nanoparticle Tracking Analysis (NTA)

The particle size and concentration were measured using a ZetaView^®^ Multiple Parameter Particle Tracking Analyzer (Particle Metrix, Inning am Ammersee, Germany) following the manufacturer’s instructions. EVs were diluted in PBS between 100- and 1000-fold to obtain a concentration within the recommended measurement range (0.5 × 10^5^ to 1010 cm^−3^). The samples were measured under the following settings: shutter: 250, sensitivity: 85, cell temperature 25 °C, 11 position, capturing under 100–200 particle per frame. Following the capture, the videos were analyzed by the built-in ZetaView Software 8.04.02 SP1 with the following parameters: maximum particle size: 800 nm, minimum size: 10 nm, minimum brightness: 22, hardware: laser >30 mW at 520 nm, camera: CMOS.

### 2.5. Western Blotting

Cells (transfected and non-transfected) were lysed by MRIPA lysis buffer (150 mM sodium chloride, 1.0% Triton X-100, 0.25% sodium deoxycholate, 50 mM Tris; pH 7.4), and the supernatant was used as cell lysates. Protein concentration was measured by Micro BCA Protein Assay kit (G Biosciences, St. Louis, MO, USA) using BSA as a standard. A total of 50 µg of the protein and 1.0 × 10^9^ EVs were denatured at 70 °C for 10 min in NuPAGE LDS Sample Buffer (Thermo Fisher Scientific, Waltham, MA, USA), separated on a 12% SDS PAGE and transferred to a nitrocellulose membrane. The membrane with the blotted proteins was blocked with blocking buffer containing 5% milk in Tris-buffered saline (TBS) with 0.1% Tween 20 (TBST) for 1 h and then incubated with a primary antibody at 4 °C overnight. Following three washes with TBST, the membrane was incubated with horseradish peroxidase-conjugated secondary antibody for 1 h at room temperature. The membrane was again washed thrice with TBST, and the protein bands were visualized by treating with SuperSignal West Pico PLUS chemiluminescent substrate (Thermo Scientific, Waltham, MA, USA); the image was captured by ChemiDoc Imaging System MP, Image Lab^TM^ Touch software version 2.2.0.08 (BioRad, Hercules, CA, USA). The following primary antibodies were used: anti-CD63 (10628D Thermo Fisher, Waltham, MA, USA), anti-TSG101 (ab125011, Abcam, Cambridge, UK), anti-ALIX (ab117600, Abcam, Cambridge, UK), anti-HA (H3663, Sigma Aldrich, St. Louis, MO, USA), and anti-calnexin (ab133615, Abcam, Cambridge, UK). The following secondary antibodies were purchased from Invitrogen (Waltham, MA, USA): goat anti-mouse IgG (H+L) highly cross-adsorbed secondary antibody, HRP (A16078), and goat anti-rabbit IgG (H+L) highly cross-adsorbed secondary antibody, HRP (A16110).

### 2.6. Immuno-Transmission Electron Microscopy (Immuno-TEM)

Carbon film-coated 200 mesh copper EM grids were soaked in 50 µL EVs (1.0 × 10^7^ naïve, E626 and RDG EVs in PBS) for 30 min for the adsorption of Evs on the grid. Evs on the grids were fixed by treating with 50 µL of 2% paraformaldehyde (PFA) for 5 min and then rinsed thrice with 100 μL PBS. To quench free aldehyde groups, we treated the grids with 50 μL of 0.05 M glycine for 10 min. The surface of the grids was blocked with a drop of blocking buffer (PBS containing 1% BSA) for 30 min. After blocking, the grids were incubated with 50 μL anti-HA (Sigma-Aldrich H3663) or anti-CD63 (Thermo Fisher 10628D) antibody (1:100 in PBS containing 0.1% BSA) for 1 h. The grids were washed five times with 50 μL PBS containing 0.1% BSA for 10 min each. For secondary antibody treatment, the grids were incubated in a drop of goat anti-mouse IgG coupled with 10 nm gold nanoparticles (Electron Microscopy Sciences, 25512) diluted at 1:100 in PBS containing 0.1% BSA for 1 h. The grids were washed five times with 50 μL PBS containing 0.1% BSA for 10 min each and then with two separate drops of (50 μL) distilled water. Evs were negatively stained with 2% uranyl acetate and then rinsed with PBS. The grids were then air dried for 24 h, and images were captured by transmission electron microscope (JEOL 1400, JEOL USA, Peabody, MA, USA) at 80 kV.

### 2.7. Bioluminescence Assay

In this assay, naïve Evs, E626-gLuc Evs, and RDG-gLuc Evs were placed in wells of a 96-well plate (UV-Star^®^ Microplate, 96 well, COC, F-Bottom Chimney Well, Greiner, Kremsmünster, Austria), uClear^®^, Clear; Greiner Bio-one, Greiner, Kremsmünster, Austria) in triplicate. A total of 95 µL of DPBS was added to each well and then the mixture was treated with 50 µL 1 µg/mL coelenterazine-H (CTZ; Regis Technologies, Morton Grove, IL, USA). The luminescence was recorded using an in vivo imaging system (IVIS; Spectrum Perkin Elmer, Aaltham, MA, USA).

The EV-binding assay was performed using bioluminescence imaging (BLI). A431 and MCF-7 cells were seeded at 2.0 × 10^4^ cells/96-well plates 24 h prior to EV treatment. The cells were treated with 2.0 × 10^7^ particles of E626-gLuc-Evs or RDG-gLuc-Evs in 100 µL media for 0, 10, 30, and 60 min at 37 °C. Following the three PBS washes to remove unbound Evs, CTZ (1 µg/mL in PBS) was added to the wells and imaged by IVIS. Total photon flux (photons/s) was quantified using Living Image 4.7.2 software (IVIS, PerkinElmer). Values are presented as the means ± SD (*n* = 4).

### 2.8. Confocal Microscopy

Co-labeled Evs were prepared from co-transfection of the plasmids as described above. A total of 2 × 10^4^ cells each of A431 and MCF-7 cells were mixed and seeded to an 8-well chamber slide (Nunc Lab-Tek, Thremo Scientific, Waltham, MA, USA) 24 h before EV treatment. These co-cultured cells were incubated with 2.0 × 10^7^ E626-mCherry or RDG-mCherry co-labeled EVs for 10 min, followed by 3× PBS washes to remove unbound EVs. The cells were fixed with 4% PFA at room temperature for 10 min, washed with PBS three times, and blocked with blocking buffer (1% BSA in PBST) for 60 min. Then, the cells were incubated with the primary antibody (CST D38B1, Cell Signaling Technology, Danvers, MA, USA) in the humidified chamber overnight at 4 °C. After three 5 min PBS washes, they were incubated in the diluted secondary antibody (CST 4412, Cell Signaling Technology, Danvers, MA, USA) for 1 h at room temperature in the dark. The slide received a coverslip following applying mounting medium (P36930 Life Technologies, Carlsbad, CA, USA) containing DAPI. Fluorescence images were taken at 60× objective magnification by confocal laser scanning microscopy (FluoView FC1000, Olympus, Shinjuku City, Tokyo, Japan).

## 3. Results

### 3.1. EV Surface Engineering Strategy and Design

First, the anti-EGFR monobody on a hydrophilic backbone was cloned into EV display construct containing lactadherin-signal peptide, monobody, 3×G4S linker, PAS40 linker, and lactadherin C1C2 domain. A flexible 3×G4S-PAS40 linker combination that improves the binding capacity in the yeast surface display system was adopted into this design (Figure 1A) [41,45]. The backbone vector (pcS) contains minimal components for cloning to keep the construct size small for improved DNA transfection efficiency. Figure 1B illustrates the steps for EV generation. HEK293T cells were transfected with the monobody–C1C2 constructs. The non-binder scrambled RDG monobody was cloned to the same backbone vector to serve as a negative control [46].

### 3.2. EV Generation and Characterization

We first generated engineered EVs from HEK293T cell culture using a generic transfection method, followed by the EV isolation from the culture media using the most widely used differential ultracentrifugation method to test the constructs and EV surface modification (Figure 1B) [44]. A key to efficient endogenous EV engineering is the high transfection efficiency of the cells, which makes HEK293T cells suitable for engineered EV generation. The transfection efficiency was confirmed by the transfection control fluorescence plasmid for each experiment. Nanoparticle tracking analysis (NTA) verified the size distribution and particle numbers for each EV type. As shown in Figure 2A, the surface modification using C1C2 fusion protein did not affect the size distribution of EVs. Western blot analysis of the bulk EV protein profiles showed enriched EV markers (CD63, TSG101, and Alix); monobody (HA); and no cell-specific marker, calnexin (Figure 2B). Immuno-transmission electron microscopy (Immuno-TEM) with anti-HA antibodies showed that the CD63-positive EVs contained E626 and RDG monobodies without notable morphological changes (Figure 2C and Appendix A). These EV characterizations are compliant with the MISEV (Minimal Information for Studies of Extracellular Vesicles) 2018 guidelines [47].

### 3.3. Engineering EVs for Imaging and Visualization

Direct EV imaging allows for an evaluation of cell-specific binding and EV uptake. As described previously, we used the endogenous loading method of EV labeling by co-transfection of fluorescence (mCherry) or bioluminescence (gaussia luciferase/gLuc) molecule–C1C2 fusion and the targeting constructs to generate dual-functional EVs [23]. The equivalent activity of imaging molecules per set of EV samples was tested prior to each experiment to avoid bias from the EV source. Appendix A shows the bioluminescence imaging of gLuc-labeled EVs and the fluorescence images of mCherry-labeled EVs, respectively. The activity of the EV imaging molecules remained stable within a week of preparation when kept at 4 °C or after a one-time freeze-thaw, possibly due to no aggregation and protein degradation.

### 3.4. In Vitro Evaluation of EV Binding

Bioluminescent co-labeled EV incubation with a live-cell culture allows for a quantitative measure of EV binding in vitro. The incubation of EGFR overexpressing A431 or MCF-7 (EGFR-) cells with 2.0 × 10^7^ of g-Luc-labeled non-tarting (RDG) or targeting (E626) EVs revealed the E626 EV accretion to A431 cells as early as 10 min, but not to MCF-7 cells (Figure 3A). In contrast, there was low or no binding of RDG EVs to MCF-7 cells or RDG EVs to either cell line, indicating that E626 monobody effectively enhanced the EV binding capacity to target cells.

Next, we examined specific binding using a direct co-culture system for confocal microscopy monitoring of EV binding using fluorescent co-labeled E626 EVs and RDG EVs (Appendix A). As shown in Figure 3B, fluorescent co-labeled E626 EVs incubated with co-culture of A431 and MCF-7 cells showed accumulation of EVs to EGFR-positive A431 cells (green), while RDG EVs did not bind to either cell type, which is consistent with the results of bioluminescence assay. Collectively, EV surface modification with high-affinity monobody renders EV’s specificity for the cell type.

## 4. Discussion

The body’s natural biomolecular transporters, extracellular vesicles (EVs), are attracting interest as drug carriers that overcome some of the issues associated with current nanoparticle delivery systems. The methods including EV engineering, synthesis, isolation, mass production, and analytical tools are evolving rapidly, yet they have considerable room for improvement and verification [48]. Targeting, one of the critical properties of cargo delivery, can be achieved by EV surface protein alteration with adhesion molecules and ligands [20,32,34,35]. Among the ligand display methods, lactadherin and Lamp2b are the two most widely used methods to date, followed by tetraspanins [20,32,35,36,37,38,49,50,51,52].

In this report, we described a method of EV surface engineering using lactadherin C1C2 gene fusion in HEK293T cells, comprehensive characterization of engineered EVs, and verification of EV binding in vitro. Previous studies demonstrated EGFR targeting using lactadherin fusion of EGFR-specific peptide [53] and nanobody [36]. The use of lactadherin C1C2 domain binding to phosphatidylserine (PS) has additional benefits, such as inhibiting the recognition of PS by coagulation factors and macrophages [36,37]. It has been shown that the purified C1C2 fusion proteins can be reconstituted with isolated EVs to engineer EV surfaces, thus potentially avoiding pDNA carryover [36,52].

HEK293 is a widely used EV-producing cell line due to the ease of gene manipulation and expansion, which has minimal effects from toxicity and immunogenicity [19]. However, due to the risk of immortalization gene packaging, such cells are excluded from a potential therapeutic EV source. In addition, while we and others have previously successfully displayed the designed peptide display on the HEK293T cell-derived EV surfaces by encoding plasmid transfection, the plasmid DNA packaging was inevitable [23,39,40,54]. Unwanted gene transcription from the strong promoter from such constructs limits their clinical applications. Nevertheless, these validations build bases to provide a proof of concept for the targeted therapy using more clinically appropriate EVs, such as fully synthetic and patient cell-derived EVs.

EV bioluminescence labeling is widely used for in vivo EV tracking due to its strong signal intensity and non-invasive nature [43,55]. In contrast, in vitro EV binding has been demonstrated using indirect labeling, such as tags, beads, and immunocapture [56,57]. Our in vitro binding assay using bioluminescence-labeled EVs showed accumulation of targeting EVs over time in a direct and quantitative manner (Figure 3A), providing a powerful tool to measure EV binding in vitro. The use of fluorescent lipid membrane dyes and membrane-permeable chemical compounds has gained popularity to visualize EV binding and uptake under a fluorescent microscope due to the ease of application [58]. However, these staining methods possibly change the biological behavior of EVs, change cellular staining patterns, or leach the dye to the cellular membrane, all of which will potentially result in artefacts or non-specific patterns [59]. The endogenous labeling method used in this report avoids such effects. Furthermore, the confocal microscopy images of the co-cultured cells with the EV accumulation validated the fact that E626 EVs bind to EGFR-overexpressing cells (Figure 3B). EVs were labeled with mCherry using endogenous mCherry-C1C2 pDNA transfection method, and fluorescence was verified prior to treatment (Appendix A). The advantage of the co-culture system binding assay is that non-EV-derived background noise is distinguishable due to the in-frame negative control, which is often a problem in insufficient signals from small EVs in the local environment. The limitations of the co-culture system include (1) co-cultured cells must have tolerable nutrient requirements, (2) long-term incubation may influence the other cells through direct contacts or secretory molecules, and (3) one needs a distinguishable cell surface marker [60].

## 5. Conclusions

We described a simple method of EV surface engineering using lactadherin C1C2 gene fusion in HEK293T cells, the comprehensive characterization of engineered EVs according to the MISEV guidelines, and the methods of EV binding analysis in vitro in detail. These methods could exploit the display of bioactive molecules, not limited to targeting and imaging on EV surfaces generated from naturally therapeutic cells or the patient’s cells for future biomedical applications.

## Figures and Tables

**Figure 1 bioengineering-09-00056-f001:**
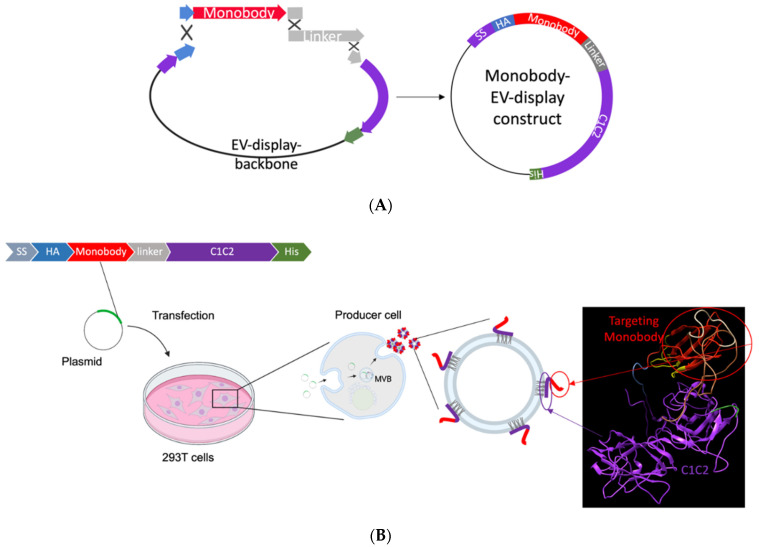
Design and schematic illustration of EV generation strategy. (**A**) The assembly cloning of the PCR fragment of the EV display vector and two synthetic double-stranded DNA fragments into a circular plasmid result in the EVs harboring encoding monobody on their surfaces. The recombinant protein composed of a signal sequence (SS), hemagglutinin tag (HA), monobody (EGFR-specific E626 or non-binder RDG), G4S-PAS linker, EV anchor region of lactadherin (C1C2), and polyhistidine tag (His). (**B**) The EV generation from HEK293T cells by transfecting the EV monobody encoding plasmid, and secretion of engineered EVs into the cultured media. (Created with BioRender.com, phyre2, and Chimera).

**Figure 2 bioengineering-09-00056-f002:**
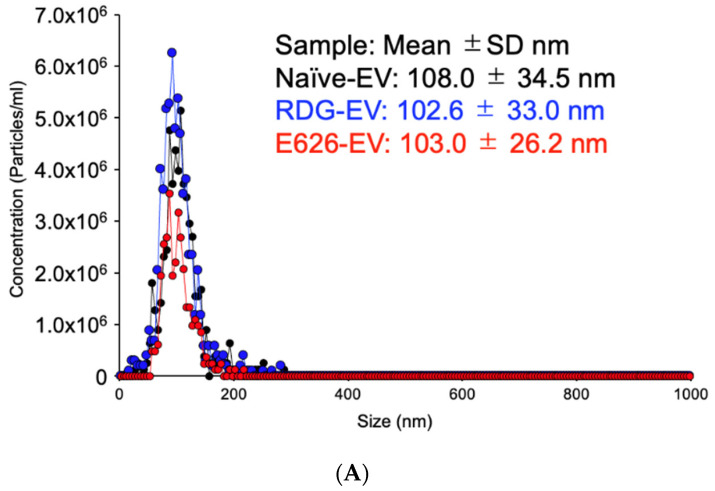
Successful isolation and characterization of engineered EVs displaying peptides of interest. (**A**) Representative size distribution of the naïve, RDG, and E626 display EVs determined by nanoparticle tracking analysis. The peak particle sizes were 106.2 nm, 98.2 nm, and 102.6 nm, respectively. (**B**) Western blot analysis of engineered EVs (RDG and E626 fusion peptide 55kDa) for the presence of EV biomarkers CD63 (30–60 kDa) and TSG101 (44 kDa), Alix (95 kDa), and peptide HA-tag. The analysis of cell lysate and engineered EVs for cellular biomarker calnexin (67KDa) is also shown. (**C**) Representative immuno-transmission electron microscopy images of naïve, RDG, and E626 EVs showing gold-labeled HA on engineered EV (RDG and E626) surfaces and CD63 EV surface markers on all the EVs.

**Figure 3 bioengineering-09-00056-f003:**
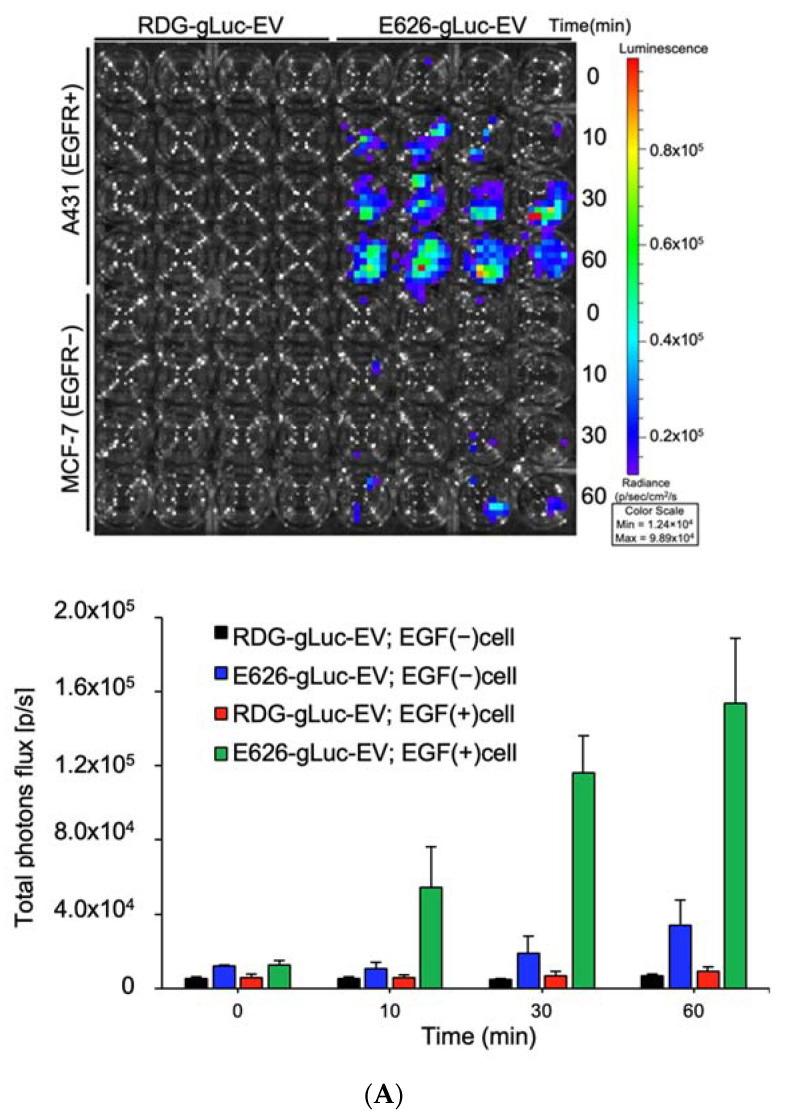
Specific binding of targeting EVs to EGFR-positive cell line in vitro. (**A**) A431 cells (EGFR-positive cells) and MCF-7 cells (EGFR-negative cells) treated with non-tarting (RDG) or targeting (E626) EVs. Representative image of EV (gLuc) binding to A431 or MCF-7 cells. The total photon flu (p/s) from EVs bound to the cells was quantified using IVIS. The value represents the means ± SD (*n* = 4) in the graph. (**B**) A431 and MCF-7 cells were co-cultured and treated either with RDG-mCherry EVs (middle row) or E626-mCherry EVs (lower row) for 10 min. The cells were fixed, and the binding was assessed by confocal laser scanning imaging of EVs (red), anti-EGFR antibody (AF488-conjugated), and nuclear staining with DAPI. microscopic images of cocultured A431 (AF488 EGFR-labeled; green), MCF-7 cells treated with EGFR-mCherry EVs, and RDG-mCherry EVs (red). DAPI stain (blue); scale bars, 20 μm.

## Data Availability

Data is contained within the article or supplementary material.

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
