# Peer review of "Design and Evaluation of Engineered Extracellular Vesicle (EV)-Based Targeting for EGFR-Overexpressing Tumor Cells Using Monobody Display"

_bioengineering, 2022, doi:10.3390/bioengineering9020056_

Round 1
Reviewer 1 Report
The article explores a new approach and tried the development of a novel robust methodology for EV engineering. As the article in point has broad implications on target-drug delivery, therefore, can revolutionize the field of therapeutics altogether.
There are many flaws in the article, although, on rectification, it can be suitable for publication.
Critical comments
- The EV-monobody display plasmid construction in line number 73 of the material and methods: This need for more descriptive in methodology even if the group has already published an article titled "Engineering Extracellular Vesicles to Target Pancreatic Tissue In Vivo" that used a similar approach, but here they haven't described it well enough for the reader to understand. You have to keep on referring to the previous article.
- Line no 75 the SLiCE assembly for monobody: The reference for this has to be changed, instead of citing a review as a reference, cite the original research article that has used this protocol. The protocol of NTA needs to be elaborated with mentioning more details about the system like the laser, the sensitivity, etc. More parameters need to be defined.
- Line no 194: The figure number is not Fig 2, but it is Fig 1 B. The manuscript has several formatting errors specifically in the fig legends.
- Line no 201 and 202 the abbreviation "HA", is used for two different terms 1) Hemagluttinin tag and polyhistidine tag. HA is quite an important marker used for monobodies in western blotting and Immuno -TEM
- EV generation and characterization: The size of the EV is determined by NTA, they need to tell that the mean size is with SD or SEM. The figure legend of Fig 2 needs to describe where need to mention of the peaks.
- Fig 2, full blot image required for CD63 cell-lysate and HA cell-lysate.
- In section 3.2, they need to mention the Fig numbers in the text as well.
8 The Immuno-TEM image for the RDG-EV with HA antibody, resolution needs to improve, it is difficult to interpret.
Author Response
Thank you very much for your reconsidering our manuscript for potential publication. We hope the new manuscript will meet your journal’s standards.
Our responses to your comment are as follows:
- The EV-monobody display plasmid construction in line number 73 of the material and methods: This need for more descriptive in methodology even if the group has already published an article titled "Engineering Extracellular Vesicles to Target Pancreatic Tissue In Vivo" that used a similar approach, but here they haven't described it well enough for the reader to understand. You have to keep on referring to the previous article.
Thank you for pointing this out. We found that our original citation number was modified by the editorial office. We fixed the citation number and included the detailed procedure.
- Line no 75 the SLiCE assembly for monobody: The reference for this has to be changed, instead of citing a review as a reference, cite the original research article that has used this protocol. The protocol of NTA needs to be elaborated with mentioning more details about the system like the laser, the sensitivity, etc. More parameters need to be defined.
Thank you for your suggestion. We changed the reference to our previous publication and added the parameters for NTA measurement.
- Line no 194: The figure number is not Fig 2, but it is Fig 1 B. The manuscript has several formatting errors specifically in the fig legends.
Thank you for catching the error. It is now fixed.
- Line no 201 and 202 the abbreviation "HA", is used for two different terms 1) Hemagluttinin tag and polyhistidine tag. HA is quite an important marker used for monobodies in western blotting and Immuno -TEM
Thank you for finding the error. The latter was a typo. We corrected it to (His) for polyhistidine tag.
- EV generation and characterization: The size of the EV is determined by NTA, they need to tell that the mean size is with SD or SEM. The figure legend of Fig 2 needs to describe where need to mention of the peaks.
Thank you for your suggestion. We included the mean sizes with SD within the figure and added the peaks in the figure legend.
- Fig 2, full blot image required for CD63 cell-lysate and HA cell-lysate.
Thank you for your suggestion. We added full blot images for CD63 and HA for cell-lysate.
- In section 3.2, they need to mention the Fig numbers in the text as well.
Thank you for your suggestion. We included the figure number in the text.
- The Immuno-TEM image for the RDG-EV with HA antibody, resolution needs to improve, it is difficult to interpret.
Thank you for your suggestion. We replaced the Immuno-TEM image for the RDG-EV-HA antibody with another one.
Reviewer 2 Report
This manuscript describes the engineering of EGFR-targeted exosomes using monobody. It is intriguing and may contribute to the advance of science in this field. However, it is required to provide in vivo proof-of-concept experiments. For example, it would be good if the authors prepare A431 tumor xenografts in mice and examine the targeting ability of the engineered exosomes using in vivo fluorescence or bioluminescence imaging.
Author Response
Thank you for the valuable feedback and suggestions.
In response to your comments, we are developing animal models to test our targeting EVs.
Our plan is to report the findings in the following publications.
Round 2
Reviewer 1 Report
The revised manuscript looks good and should be accepted
Reviewer 2 Report
I have no further comments regarding this manuscript. It would be fine to be published in this journal.